# Recurrent Relational Networks

**Rasmus Berg Palm**
Technical University of Denmark
Tradeshift
rapal@dtu.dk

**Ulrich Paquet**
DeepMind
upaq@google.com

**Ole Winther**
Technical University of Denmark
olwi@dtu.dk

## Abstract

This paper is concerned with learning to solve tasks that require a chain of interdependent steps of relational inference, like answering complex questions about the relationships between objects, or solving puzzles where the smaller elements of a solution mutually constrain each other. We introduce the *recurrent relational network*, a general purpose module that operates on a *graph* representation of objects. As a generalization of Santoro et al. [2017]'s relational network, it can augment any neural network model with the capacity to do many-step relational reasoning. We achieve state of the art results on the bAbI textual question-answering dataset with the recurrent relational network, consistently solving 20/20 tasks. As bAbI is not particularly challenging from a relational reasoning point of view, we introduce Pretty-CLEVR, a new diagnostic dataset for relational reasoning. In the Pretty-CLEVR set-up, we can vary the question to control for the number of relational reasoning steps that are required to obtain the answer. Using Pretty-CLEVR, we probe the limitations of multi-layer perceptrons, relational and recurrent relational networks. Finally, we show how recurrent relational networks can learn to solve Sudoku puzzles from supervised training data, a challenging task requiring upwards of 64 steps of relational reasoning. We achieve state-of-the-art results amongst comparable methods by solving 96.6% of the hardest Sudoku puzzles.

## 1   Introduction

A central component of human intelligence is the ability to abstractly reason about objects and their interactions [Spelke et al., 1995, Spelke and Kinzler, 2007]. As an illustrative example, consider solving a Sudoku. A Sudoku consists of 81 cells that are arranged in a 9-by-9 grid, which must be filled with digits 1 to 9 so that each digit appears exactly once in each row, column and 3-by-3 non-overlapping box, with a number of digits given [1]. To solve a Sudoku, one methodically reasons about the puzzle in terms of its cells and their interactions over many steps. One tries placing digits in cells and see how that affects other cells, iteratively working toward a solution.

Contrast this with the canonical deep learning approach to solving problems, the multilayer perceptron (MLP), or multilayer convolutional neural net (CNN). These architectures take the entire Sudoku as an input and output the entire solution in a single forward pass, ignoring the inductive bias that objects exists in the world, and that they affect each other in a consistent manner. Not surprisingly these models fall short when faced with problems that require even basic relational reasoning [Lake et al., 2016, Santoro et al., 2017].

The relational network of Santoro et al. [2017] is an important first step towards a simple module for reasoning about objects and their interactions but it is limited to performing a single relational operation, and was evaluated on datasets that require a maximum of three steps of reasoning (which,

surprisingly, can be solved by a single relational reasoning step as we show). Looking beyond relational networks, there is a rich literature on logic and reasoning in artificial intelligence and machine learning, which we discuss in section 5.

Toward generally realizing the ability to methodically reason about objects and their interactions over many steps, this paper introduces a composite function, the *recurrent relational network*. It serves as a modular component for many-step relational reasoning in end-to-end differentiable learning systems. It encodes the inductive biases that 1) objects exists in the world 2) they can be sufficiently described by properties 3) properties can change over time 4) objects can affect each other and 5) given the properties, the effects object have on each other is *invariant to time*.

An important insight from the work of Santoro et al. [2017] is to decompose a function for relational reasoning into two components or "modules": a perceptual front-end, which is tasked to recognize objects in the raw input and represent them as vectors, and a relational reasoning module, which uses the representation to reason about the objects and their interactions. Both modules are trained jointly end-to-end. In computer science parlance, the relational reasoning module implements an *interface*: it operates on a graph of nodes and directed edges, where the nodes are represented by real valued vectors, and is differentiable. This paper chiefly develops the relational reasoning side of that interface.

Some of the tasks we evaluate on can be efficiently and perfectly solved by hand-crafted algorithms that operate on the symbolic level. For example, 9-by-9 Sudokus can be solved in a fraction of a second with constraint propagation and search [Norvig, 2006] or with dancing links [Knuth, 2000]. These symbolic algorithms are superior in every respect but one: they don't comply with the interface, as they are not differentiable and don't work with real-valued vector descriptions. They therefore cannot be used in a combined model with a deep learning perceptual front-end and learned end-to-end.

Following Santoro et al. [2017], we use the term "relational reasoning" liberally for an object- and interaction-centric approach to problem solving. Although the term "relational reasoning" is similar to terms in other branches of science, like relational logic or first order logic, no direct parallel is intended.

This paper considers many-step relational reasoning, a challenging task for deep learning architectures. We develop a recurrent relational reasoning module, which constitutes our main contribution. We show that it is a powerful architecture for many-step relational reasoning on three varied datasets, achieving state-of-the-art results on bAbI and Sudoku.

## 2 Recurrent Relational Networks

We ground the discussion of a recurrent relational network in something familiar, solving a Sudoku puzzle. A simple strategy works by noting that if a certain Sudoku cell is given as a "7", one can safely remove "7" as an option from other cells in the same row, column and box. In a message passing framework, that cell needs to send a message to each other cell in the same row, column, and box, broadcasting it's value as "7", and informing those cells not to take the value "7". In an iteration $t$, these messages are sent simultaneously, in parallel, between all cells. Each cell $i$ should then consider all incoming messages, and update its internal state $h_i^t$ to $h_i^{t+1}$. With the updated state each cell should send out new messages, and the process repeats.

**Message passing on a graph.** The recurrent relational network will learn to pass messages on a *graph*. For Sudoku, the graph has $i \in \{1, 2, ..., 81\}$ nodes, one for each cell in the Sudoku. Each node has an input feature vector $x_i$, and edges to and from all nodes that are in the same row, column and box in the Sudoku. The *graph* is the input to the relational reasoning module, and vectors $x_i$ would generally be the output of a perceptual front-end, for instance a convolutional neural network. Keeping with our Sudoku example, each $x_i$ encodes the initial cell content (empty or given) and the row and column position of the cell.

At each step $t$ each node has a hidden state vector $h_i^t$, which is initialized to the features, such that $h_i^0 = x_i$. At each step $t$, each node sends a message to each of its neighboring nodes. We define the message $m_{ij}^t$ from node $i$ to node $j$ at step $t$ by

$$m_{ij}^t = f\left(h_i^{t-1}, h_j^{t-1}\right) , \tag{1}$$

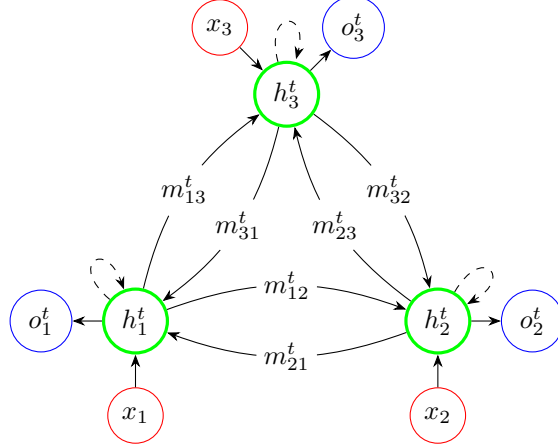

Figure 1: A *recurrent relational network* on a fully connected graph with 3 nodes. The nodes' hidden states $h_i^t$ are highlighted with green, the inputs $x_i$ with red, and the outputs $o_i^t$ with blue. The dashed lines indicate the recurrent connections. Subscripts denote node indices and superscripts denote steps $t$. For a figure of the same graph unrolled over 2 steps see the supplementary material.

where $f$, the message function, is a multi-layer perceptron. This allows the network to learn what kind of messages to send. In our experiments, MLPs with linear outputs were used. Since a node needs to consider all the incoming messages we sum them with

$$m_j^t = \sum_{i \in N(j)} m_{ij}^t \, , \tag{2}$$

where $N(j)$ are all the nodes that have an edge into node $j$. For Sudoku, $N(j)$ contains the nodes in the same row, column and box as $j$. In our experiments, since the messages in (1) are linear, this is similar to how log-probabilities are summed in belief propagation [Murphy et al., 1999].

**Recurrent node updates.**　Finally we update the node hidden state via

$$h_j^t = g\left(h_j^{t-1}, x_j, m_j^t\right) \, , \tag{3}$$

where $g$, the node function, is another learned neural network. The dependence on the previous node hidden state $h_j^{t-1}$ allows the network to iteratively work towards a solution instead of starting with a blank slate at every step. Injecting the feature vector $x_j$ at each step like this allows the node function to focus on the messages from the other nodes instead of trying to remember the input.

**Supervised training.**　The above equations for sending messages and updating node states define a recurrent relational network's core. To train a recurrent relational network in a supervised manner to solve a Sudoku we introduce an output probability distribution over the digits 1-9 for each of the nodes in the graph. The output distribution $o_i^t$ for node $i$ at step $t$ is given by

$$o_i^t = r\left(h_i^t\right) \, , \tag{4}$$

where $r$ is a MLP that maps the node hidden state to the output probabilities, e.g. using a softmax nonlinearity. Given the target digits $\mathbf{y} = \{y_1, y_2, ..., y_{81}\}$ the loss at step $t$, is then the sum of cross-entropy terms, one for each node: $l^t = -\sum_{i=1}^{I} \log o_i^t[y_i]$, where $o_i[y_i]$ is the $y_i$'th component of $o_i$. Equations (1) to (4) are illustrated in figure 1.

**Convergent message passing.**　A distinctive feature of our proposed model is that we minimize the cross entropy between the output and target distributions at *every step*.

At test time we only consider the output probabilities at the last step, but having a loss at every step during training is beneficial. Since the target digits $y_i$ are constant over the steps, it encourages the network to learn a convergent message passing algorithm. Secondly, it helps with the vanishing gradient problem.

**Variations.**   If the edges are unknown, the graph can be assumed to be fully connected. In this case the network will need to learn which objects interact with each other. If the edges have attributes, $e_{ij}$, the message function in equation 1 can be modified such that $m_{ij}^t = f\left(h_i^{t-1}, h_j^{t-1}, e_{ij}\right)$. If the output of interest is for the whole graph instead of for each node the output in equation 4 can be modified such that there's a single output $o^t = r\left(\sum_i h_i^t\right)$. The loss can be modified accordingly.

## 3   Experiments

Code to reproduce all experiments can be found at github.com/rasmusbergpalm/recurrent-relational-networks.

### 3.1   bAbI question-answering tasks

Table 1: bAbI results. Trained jointly on all 20 tasks using the 10,000 training samples. Entries marked with an asterix are our own experiments, the rest are from the respective papers.

| Method | N | Mean Error (%) | Failed tasks (err. >5%) |
|---|---|---|---|
| *RRN** (this work) | 15 | **0.46 ± 0.77** | **0.13 ± 0.35** |
| SDNC [Rae et al., 2016] | 15 | 6.4 ± 2.5 | 4.1 ± 1.6 |
| DAM [Rae et al., 2016] | 15 | 8.7 ± 6.4 | 5.4 ± 3.4 |
| SAM [Rae et al., 2016] | 15 | 11.5 ± 5.9 | 7.1 ± 3.4 |
| DNC [Rae et al., 2016] | 15 | 12.8 ± 4.7 | 8.2 ± 2.5 |
| NTM [Rae et al., 2016] | 15 | 26.6 ± 3.7 | 15.5 ± 1.7 |
| LSTM [Rae et al., 2016] | 15 | 28.7 ± 0.5 | 17.1 ± 0.8 |
| EntNet [Henaff et al., 2016] | 5 | 9.7 ± 2.6 | 5 ± 1.2 |
| ReMo [Yang et al., 2018] | 1 | 1.2 | 1 |
| RN [Santoro et al., 2017] | 1 | N/A | 2 |
| MemN2N [Sukhbaatar et al., 2015] | 1 | 7.5 | 6 |

bAbI is a text based QA dataset from Facebook [Weston et al., 2015] designed as a set of prerequisite tasks for reasoning. It consists of 20 types of tasks, with 10,000 questions each, including deduction, induction, spatial and temporal reasoning. Each question, e.g. "Where is the milk?" is preceded by a number of facts in the form of short sentences, e.g. "Daniel journeyed to the garden. Daniel put down the milk." The target is a single word, in this case "garden", one-hot encoded over the full bAbI vocabulary of 177 words. A task is considered solved if a model achieves greater than 95% accuracy. The most difficult tasks require reasoning about three facts.

To map the questions into a graph we treat the facts related to a question as the nodes in a fully connected graph up to a maximum of the last 20 facts. The fact and question sentences are both encoded by Long Short Term Memory (LSTM) [Hochreiter and Schmidhuber, 1997] layers with 32 hidden units each. We concatenate the last hidden state of each LSTM and pass that through a MLP. The output is considered the node features $x_i$. Following [Santoro et al., 2017] all edge features $e_{ij}$ are set to the question encoding. We train the network for three steps. At each step, we sum the node hidden states and pass that through a MLP to get a single output for the whole graph. For details see the supplementary material.

Our trained network solves 20 of 20 tasks in 13 out of 15 runs. This is state-of-the-art and markedly more stable than competing methods. See table 1. We perform ablation experiment to see which parts of the model are important, including varying the number of steps. We find that using dropout and appending the question encoding to the fact encodings is important for the performance. See the supplementary material for details.

Surprisingly, we find that we only need a single step of relational reasoning to solve all the bAbI tasks. This is surprising since the hardest tasks requires reasoning about three facts. It's possible that there are superficial correlations in the tasks that the model learns to exploit. Alternatively the model learns to compress all the relevant fact-relations into the 128 floats resulting from the sum over the node hidden states, and perform the remaining reasoning steps in the output MLP. Regardless, it appears multiple steps of relational reasoning are not important for the bAbI dataset.

## 3.2 Pretty-CLEVR

Given that bAbI did not require multiple steps of relational reasoning and in order to test our hypothesis that our proposed model is better suited for tasks requiring more steps of relational reasoning we create a diagnostic dataset "Pretty-CLEVER". It can be seen as an extension of the "Sort-of-CLEVR" data set by [Santoro et al., 2017] which has questions of a non-relational and relational nature. "Pretty-CLEVR" takes this a step further and has non-relational questions as well as questions requiring *varying* degrees of relational reasoning.

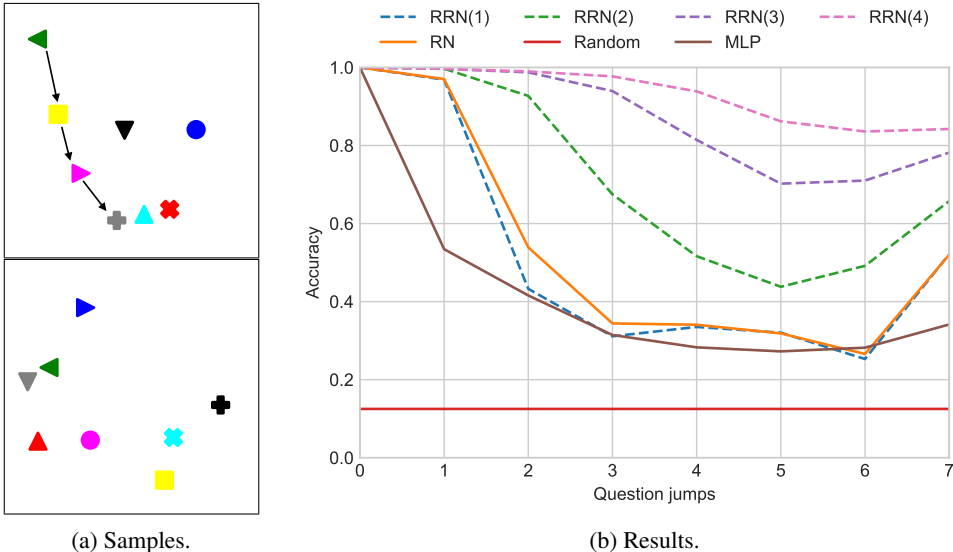

| (a) Samples. | (b) Results. |

Figure 2: 2a Two samples of the Pretty-CLEVR diagnostic dataset. Each sample has 128 questions associated, exhibiting varying levels of relational reasoning difficulty. For the topmost sample the solution to the question: "green, 3 jumps", which is "plus", is shown with arrows. 2b Random corresponds to picking one of the eight possible outputs at random (colors or shapes, depending on the input). The RRN is trained for four steps but since it predicts at each step we can evaluate the performance for each step. The the number of steps is stated in parentheses.

Pretty-CLEVR consists of scenes with eight colored shapes and associated questions. Questions are of the form: "Starting at object X which object is N jumps away?". Objects are uniquely defined by their color or shape. If the start object is defined by color, the answer is a shape, and vice versa. Jumps are defined as moving to the closest object, without going to an object already visited. See figure 2a. Questions with zero jumps are non-relational and correspond to: "What color is shape X?" or "What shape is color X?". We create 100,000 random scenes, and 128 questions for each (8 start objects, 0-7 jumps, output is color or shape), resulting in 12.8M questions. We also render the scenes as images. The "jump to nearest" type question is chosen in an effort to eliminate simple correlations between the scene state and the answer. It is highly non-linear in the sense that slight differences in the distance between objects can cause the answer to change drastically. It is also asymmetrical, i.e. if the question "x, n jumps" equals "y", there is no guarantee that "y, n jumps" equals "x". We find it is a surprisingly difficult task to solve, even with a powerful model such as the RRN. We hope others will use it to evaluate their relational models.[2]

Since we are solely interested in examining the effect of multiple steps of relational reasoning we train on the state descriptions of the scene. We consider each scene as a fully connected undirected graph with 8 nodes. The feature vector for each object consists of the position, shape and color. We encode the question as the start object shape or color and the number of jumps. As we did for bAbI we concatenate the question and object features and pass it through a MLP to get the node features $x_i$. To make the task easier we set the edge features to the euclidean distance between the objects. We train our network for four steps and compare to a single step relational network and a baseline

MLP that considers the entire scene state, all pairwise distances, and the question as a single vector. For details see the supplementary material.

Mirroring the results from the "Sort-of-CLEVR" dataset the MLP perfectly solves the non-relational questions, but struggle with even single jump questions and seem to lower bound the performance of the relational networks. The relational network solves the non-relational questions as well as the ones requiring a single jump, but the accuracy sharply drops off with more jumps. This matches the performance of the recurrent relational network which generally performs well as long as the number of steps is greater than or equal to the number of jumps. See fig 2b. It seems that, despite our best efforts, there are spurious correlations in the data such that questions with six to seven jumps are easier to solve than those with four to five jumps.

### 3.3 Sudoku

We create training, validation and testing sets totaling 216,000 Sudoku puzzles with a uniform distribution of givens between 17 and 34. We consider each of the 81 cells in the 9x9 Sudoku grid a node in a graph, with edges to and from each other cell in the same row, column and box. The node features $x_i$ are the output of a MLP which takes as input the digit for the cell (0-9, 0 if not given), and the row and column position (1-9). Edge features are not used. We run the network for 32 steps and at every step the output function $r$ maps each node hidden state to nine output logits corresponding to the nine possible digits. For details see the supplementary material.

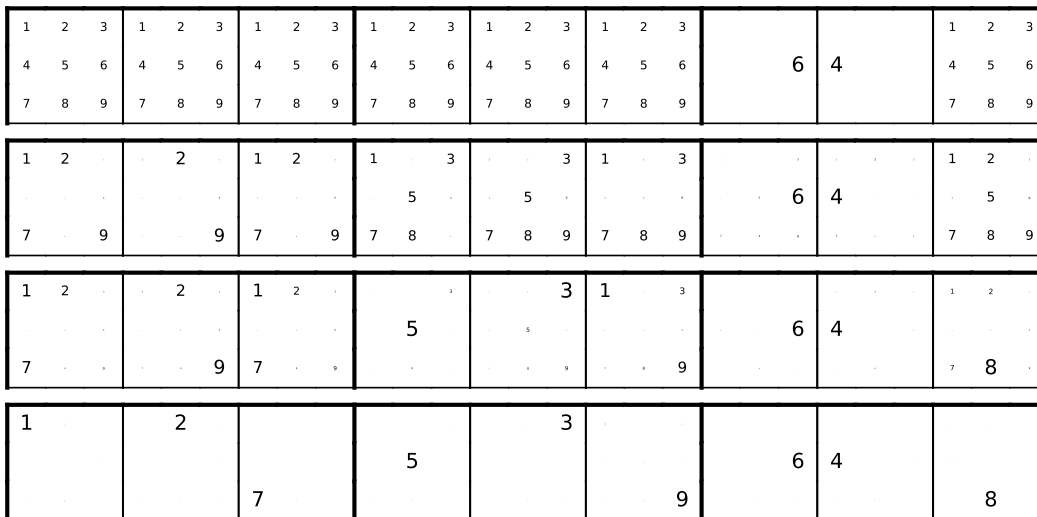

Figure 3: Example of how the trained network solves part of a Sudoku. Only the top row of a full 9x9 Sudoku is shown for clarity. From top to bottom steps 0, 1, 8 and 24 are shown. See the supplementary material for a full Sudoku. Each cell displays the digits 1-9 with the font size scaled (non-linearly for legibility) to the probability the network assigns to each digit. Notice how the network eliminates the given digits 6 and 4 from the other cells in the first step. Animations showing how the trained network solves Sodukos, including a failure case can be found at imgur.com/a/ALsfB.

Our network learns to solve 94.1% of even the hardest 17-givens Sudokus after 32 steps. We only consider a puzzled solved if all the digits are correct, i.e. no partial credit is given for getting individual digits correct. For more givens the accuracy (fraction of test puzzles solved) quickly approaches 100%. Since the network outputs a probability distribution for each step, we can visualize how the network arrives at the solution step by step. For an example of this see figure 3.

To examine our hypothesis that multiple steps are required we plot the accuracy as a function of the number of steps. See figure 4. We can see that even simple Sudokus with 33 givens require upwards of 10 steps of relational reasoning, whereas the harder 17 givens continue to improve even after 32 steps. Figure 4 also shows that the model has learned a convergent algorithm. The model was trained for 32 steps, but seeing that the accuracy increased with more steps, we ran the model for 64 steps during testing. At 64 steps the accuracy for the 17 givens puzzles increases to 96.6%.

We also examined the importance of the row and column features by multiplying the row and column embeddings by zero and re-tested our trained network. At 64 steps with 17 givens, the accuracy changed to 96.7%. It thus seems the network does not use the row and column position information to solve the task.

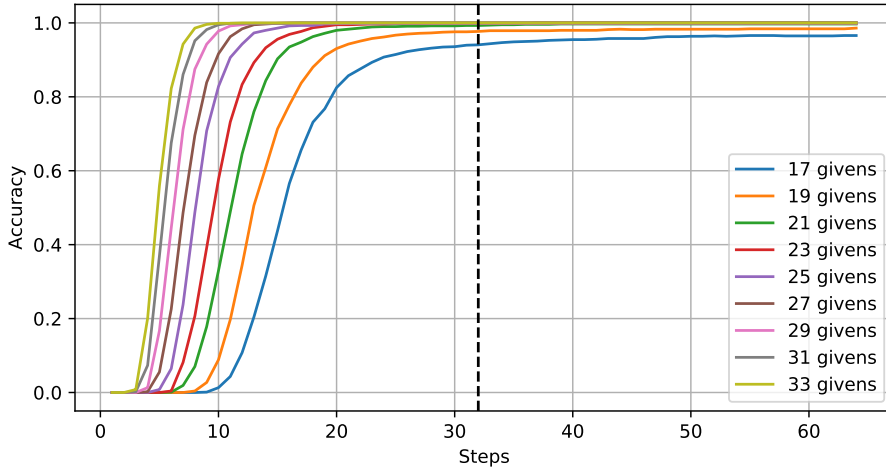

Figure 4: Fraction of test puzzles solved as a function of number of steps. Even simple Sudokus with 33 givens require about 10 steps of relational reasoning to be solved. The dashed vertical line indicates the 32 steps the network was trained for. The network appears to have learned a convergent relational reasoning algorithm such that more steps beyond 32 improve on the hardest Sudokus.

We compare our network to several other differentiable methods. See table 2. We train two relational networks: a node and a graph centric. For details see the supplementary material. Of the two, the node centric was considerably better. The node centric correspond exactly to our proposed network with a single step, yet fails to solve any Sudoku. This shows that multiple steps are crucial for complex relational reasoning. Our network outperforms loopy belief propagation, with parallel and random messages passing updates [Bauke, 2008]. It also outperforms a version of loopy belief propagation modified specifically for solving Sudokus that uses 250 steps, Sinkhorn balancing every two steps and iteratively picks the most probable digit [Khan et al., 2014]. We also compare to learning the messages in parallel loopy BP as presented in Lin et al. [2015]. We tried a few variants including a single step as presented and 32 steps with and without a loss on every step, but could not get it to solve any 17 given Sudoku. Finally we outperform Park [2016] which treats the Sudoku as a 9x9 image, uses 10 convolutional layers, iteratively picks the most probable digit, and evaluate on easier Sudokus with 24-36 givens. We also tried to train a version of our network that only had a loss at the last step. It was harder to train, performed worse and didn't learn a convergent algorithm.

Table 2: Comparison of methods for solving Sudoku puzzles. Only methods that are differentiable are included in the comparison. Entries marked with an asterix are our own experiments, the rest are from the respective papers.

| Method | Givens | Accuracy |
| --- | --- | --- |
| *Recurrent Relational Network* (this work) | 17 | **96.6%** |
| Loopy BP, modified [Khan et al., 2014] | 17 | 92.5% |
| Loopy BP, random [Bauke, 2008] | 17 | 61.7% |
| Loopy BP, parallel [Bauke, 2008] | 17 | 53.2% |
| Deeply Learned Messages* [Lin et al., 2015] | 17 | 0% |
| Relational Network, node* [Santoro et al., 2017] | 17 | 0% |
| Relational Network, graph* [Santoro et al., 2017] | 17 | 0% |
| Deep Convolutional Network [Park, 2016] | 24-36 | 70% |

### 3.4 Age arithmetic

Anonymous reviewer 2 suggested the following task which we include here. The task is to infer the age of a person given a single absolute age and a set of age differences, e.g. "Alice is 20 years old. Alice is 4 years older than Bob. Charlie is 6 years younger than Bob. How old is Charlie?". Please see the supplementary material for details on the task and results.

## 4 Discussion

We have proposed a general relational reasoning model for solving tasks requiring an order of magnitude more complex relational reasoning than the current state-of-the art. BaBi and Sort-of-CLEVR require a few steps, Pretty-CLEVR requires up to eight steps and Sudoku requires more than ten steps. Our relational reasoning module can be added to any deep learning model to add a powerful relational reasoning capacity. We get state-of-the-art results on Sudokus solving 96.6% of the hardest Sudokus with 17 givens. We also markedly improve state-of-the-art on the BaBi dataset solving 20/20 tasks in 13 out of 15 runs with a single model trained jointly on all tasks.

One potential issue with having a loss at every step is that it might encourage the network to learn a greedy algorithm that gets stuck in a local minima. However, the output function $r$ separates the node hidden states and messages from the output probability distributions. The network therefore has the capacity to use a small part of the hidden state for retaining a current best guess, which can remain constant over several steps, and other parts of the hidden state for running a non-greedy multi-step algorithm.

Sending messages for all nodes in parallel and summing all the incoming messages might seem like an unsophisticated approach that risk resulting in oscillatory behavior and drowning out the important messages. However, since the receiving node hidden state is an input to the message function, the receiving node can in a sense determine which messages it wishes to receive. As such, the sum can be seen as an implicit attention mechanism over the incoming messages. Similarly the network can learn an optimal message passing schedule, by ignoring messages based on the history and current state of the receiving and sending node.

## 5 Related work

Relational networks [Santoro et al., 2017] and interaction networks [Battaglia et al., 2016] are the most directly comparable to ours. These models correspond to using a single step of equation 3. Since it only does one step it cannot naturally do complex multi-step relational reasoning. In order to solve the tasks that require more than a single step it must compress all the relevant relations into a fixed size vector, then perform the remaining relational reasoning in the last forward layers. Relational networks, interaction networks and our proposed model can all be seen as an instance of Graph Neural Networks [Scarselli et al., 2009, Gilmer et al., 2017].

Graph neural networks with message passing computations go back to Scarselli et al. [2009]. However, there are key differences that we found important for implementing stable multi-step relational reasoning. Including the node features $x_j$ at every step in eq. 3 is important to the stability of the network. Scarselli et al. [2009], eq. 3 has the node features, $l_n$, inside the message function. Battaglia et al. [2016] use an $x_j$ in the node update function, but this is an external driving force. Sukhbaatar et al. [2016] also proposed to include the node features at every step. Optimizing the loss at every step in order to learn a convergent message passing algorithm is novel to the best of our knowledge. Scarselli et al. [2009] introduces an explicit loss term to ensure convergence. Ross et al. [2011] trains the inference machine predictors on every step, but there are no hidden states; the node states are the output marginals directly, similar to how belief propagation works.

Our model can also be seen as a completely learned message passing algorithm. Belief propagation is a hand-crafted message passing algorithm for performing exact inference in directed acyclic graphical models. If the graph has cycles, one can use a variant, loopy belief propagation, but it is not guaranteed to be exact, unbiased or converge. Empirically it works well though and it is widely used [Murphy et al., 1999]. Several works have proposed replacing parts of belief propagation with learned modules [Heess et al., 2013, Lin et al., 2015]. Our work differs by not being rooted in loopy BP, and instead learning all parts of a general message passing algorithm. Ross et al. [2011] proposes

Inference Machines which ditch the belief propagation algorithm altogether and instead train a series of regressors to output the correct marginals by passing messages on a graph. Wei et al. [2016] applies this idea to pose estimation using a series of convolutional layers and Deng et al. [2016] introduces a recurrent node update for the same domain.

There is rich literature on combining symbolic reasoning and logic with sub-symbolic distributed representations which goes all the way back to the birth of the idea of parallel distributed processing McCulloch and Pitts [1943]. See [Raedt et al., 2016, Besold et al., 2017] for two recent surveys. Here we describe only a few recent methods. Serafini and Garcez [2016] introduces the Logic Tensor Network (LTN) which describes a first order logic in which symbols are grounded as vector embeddings, and predicates and functions are grounded as tensor networks. The embeddings and tensor networks are then optimized jointly to maximize a fuzzy satisfiability measure over a set of known facts and fuzzy constraints. Šourek et al. [2015] introduces the Lifted Relational Network which combines relational logic with neural networks by creating neural networks from lifted rules and training examples, such that the connections between neurons created from the same lifted rules shares weights. Our approach differs fundamentally in that we do not aim to bridge symbolic and sub-symbolic methods. Instead we stay completely in the sub-symbolic realm. We do not introduce or consider any explicit logic, aim to discover (fuzzy) logic rules, or attempt to include prior knowledge in the form of logical constraints.

Amos and Kolter [2017] Introduces OptNet, a neural network layer that solve quadratic programs using an efficient differentiable solver. OptNet is trained to solve 4x4 Sudokus amongst other problems and beats the deep convolutional network baseline as described in Park [2016]. Unfortunately we cannot compare to OptNet directly as it has computational issues scaling to 9x9 Sudokus (Brandon Amos, 2018, personal communication).

Sukhbaatar et al. [2016] proposes the Communication Network (CommNet) for learning multi-agent cooperation and communication using back-propagation. It is similar to our recurrent relational network, but differs in key aspects. The messages passed between all nodes at a given step are the same, corresponding to the average of all the node hidden states. Also, it is not trained to minimize the loss on every step of the algorithm.

**Acknowledgments**

We'd like to thank the anonymous reviewers for the valuable comments and suggestions, especially reviewer 2 who suggested the age arithmetic task. This research was supported by the NVIDIA Corporation with the donation of TITAN X GPUs.

## Footnotes

[1]We invite the reader to solve the Sudoku in the supplementary material to appreciate the difficulty of solving a Sudoku in which 17 cells are initially filled.

[2]Pretty-CLEVR is available online as part of the code for reproducing experiments.

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
