[Supplementary Material]

# Recurrent Relational Networks

**Rasmus Berg Palm**
Technical University of Denmark
Tradeshift
rapal@dtu.dk

**Ulrich Paquet**
DeepMind
upaq@google.com

**Ole Winther**
Technical University of Denmark
olwi@dtu.dk

## 1 Supplementary Material

### 1.1 bAbI experimental details

Unless otherwise specified we use 128 hidden units for all layers and all MLPs are 3 ReLU layers followed by a linear layer.

We compute each node feature vector as

$$x_i = \text{MLP}(\text{concat}(\text{last}(\text{LSTM}_S(s_i)), \text{last}(\text{LSTM}_Q(q)), \text{onehot}(p_i + o)))$$

where $s_i$ is fact $i$, $q$ is the question, $p_i$ is the sentence position (1-20) of fact $i$ and $o$ is a random offset per question (1-20), such that the onehot output is 40 dimensional. The offset is constant for all facts related to a single question to avoid changing the relative order of the facts. The random offset prevents the network from memorizing the position of the facts and rather reason about their ordering. Our message function $f$ is a MLP. Our node function $g$ uses an LSTM over reasoning steps

$$h_j^t, s_j^t = \text{LSTM}_G(\text{MLP}(\text{concat}(x_j, m_j^t)), s_j^{t-1}) \,,$$

where $s_j^t$ is the cell state of the LSTM for unit $j$ at time $t$. $s_j^0$ is initialized to zero.

We run our network for three steps. To get a graph level output, we use a MLP over the sum of the node hidden states, $o^t = \text{MLP}\left(\sum_i h_i^t\right)$ with 3 layers, the final being a linear layer that maps to the output logits. The last two layers uses dropout of 50%. We train and validate on all 20 tasks jointly using the 9,000 training and 1,000 validation samples defined in the `en_valid_10k` split. We use the Adam optimizer with a batch size of 512, a learning rate of 2e-4 and L2 regularization with a rate of 1e-5. We train for 5M gradient steps.

### 1.2 bAbI ablation experiments

To test which parts of the proposed model is important to solving the bAbI tasks we perform ablation experiments. One of the main differences between the relational network and our proposed model, aside from the recurrent steps, is that we encode the sentences and question together. We ablate the model in two ways to test how important this is. 1) Using a single linear layer instead of the 4-layer MLP baseline, and 2) Not encoding them together. In this case the node hidden states are initialized to the fact encodings. We found dropout to be important, so we also perform an ablation experiment without dropout. We run each ablation experiment eight times. We also do pseudo-ablation experiments with fewer steps by measuring at each step of the RRN. See table 1.

### 1.3 Pretty-CLEVR experimental details

Our setup for Pretty-CLEVR is a bit simpler than for bAbI. Unless otherwise specified we use 128 hidden units for all hidden layers and all MLPs are 1 ReLU layer followed by a linear layer.

| Model | Runs | Mean Error (%) | Failed tasks (err. >5%) | Mean error @ 1M updates (%) |
|---|---|---|---|---|
| Baseline, 3 steps | 15 | $0.46 \pm 0.77$ | $0.13 \pm 0.35$ | $1.83 \pm 1.06$ |
| Baseline, 2 steps | 15 | $0.46 \pm 0.76$ | $0.13 \pm 0.35$ | $1.83 \pm 1.06$ |
| Baseline, 1 step | 15 | $0.48 \pm 0.79$ | $0.13 \pm 0.35$ | $1.84 \pm 1.06$ |
| linear encoding | 8 | $\mathbf{0.20 \pm 0.01}$ | $\mathbf{0 \pm 0}$ | $\mathbf{0.63 \pm 0.69}$ |
| no encoding | 8 | $0.53 \pm 0.91$ | $0.13 \pm 0.35$ | $2.39 \pm 1.73$ |
| no dropout | 8 | $1.74 \pm 1.28$ | $0.63 \pm 0.52$ | $2.57 \pm 0.95$ |

Table 1: BaBi ablation results.

We compute each node feature vector $x_i$ as

$$o_i = \text{concat}(p_i, \text{onehot}(c_i), \text{onehot}(m_i))$$
$$q = \text{concat}(\text{onehot}(s), \text{onehot}(n))$$
$$x_i = \text{MLP}(\text{concat}(o_i, q))$$

where $p_i \in [0,1]^2$ is the position, $\mathbb{N}^n \equiv \{0, ..., n-1\}$, $c_i \in \mathbb{N}^8$ is the color, $m_i \in \mathbb{N}^8$ is the marker, $s \in \mathbb{N}^{16}$ is the marker or color of the start object, and $n \in \mathbb{N}^8$ is the number of jumps.

Our message function $f$ is a MLP. Our node function $g$ is,

$$h_j^t = \text{MLP}(\text{concat}(h_j^{t-1}, x_j, m_j^t))$$

Our output function $r$ is a MLP with a dropout fraction of 0.5 in the penultimate layer. The last layer has 16 hidden linear units. We run our recurrent relational network for 4 steps.

We train on the 12.8M training questions, and augment the data by scaling and rotating the scenes randomly. We use separate validation and test sets of 128.000 questions each. We use the Adam optimizer with a learning rate of 1e-4 and train for 10M gradient updates with a batch size of 128.

The baseline RN is identical to the described RRN, except it only does a single step of relational reasoning.

The baseline MLP takes the entire scene state, $\mathbf{x}$, as an input, such that

$$\mathbf{x} = \text{concat}(o_0, ..., o_7, d_{00}, ..., d_{77}, q)$$

where $d_{ij} \in \mathbb{R}$ is the euclidean distance from object $i$ to $j$.

The baseline MLP has 4 ReLu layers with 256 hidden units, with dropout of 0.5 on the last layer, followed by a linear layer with 16 hidden units. The baseline MLP has 87% more parameters than the RRN and RN (261,136 vs 139,536).

### 1.4 Sudoku dataset

To generate our dataset the starting point is the collection of 49,151 unique 17-givens puzzles gathered by Royle [2014] which we solve using the solver from Norvig [2006]. Then we split the puzzles into a test, validation and training *pool*, with 10,000, 1,000 and 38,151 samples respectively. To generate the *sets* we train, validate and test on we do the following: for each $n \in \{0, ..., 17\}$ we sample $k$ puzzles from the respective pool, with replacement. For each sampled puzzle we add $n$ random digits from the solution. We then swap the digits according to a random permutation, e.g. $1 \rightarrow 5$, $2 \rightarrow 3$, etc. The resulting puzzle is added to the respective set. For the test, validation and training sets we sample $k = 1,000$, $k = 1,000$ and $k = 10,000$ puzzles in this way.

### 1.5 Sudoku experimental details

Unless otherwise specified we use 96 hidden units for all hidden layers and all MLPs are 3 ReLU layers followed by a linear layer.

Denote the digit for cell $j$ $d_j$ (0-9, 0 if not given), and the row and column position $\text{row}_j$ (1-9) and $\text{column}_j$ (1-9) respectively.. The node features are then

$$x_j = \text{MLP}(\text{concat}(\text{embed}(d_j), \text{embed}(\text{row}_j), \text{embed}(\text{column}_j)))$$

where each embed is a 16 dimensional learned embedding. We could probably have used one-hot encoding instead of the embeddings, embedding was just the first thing we tried. Edge features were not used. The message function $f$ is an MLP. The node function $g$, is identical to the setup for bAbI, i.e.

$$h_j^t, s_j^t = \text{LSTM}_G(\text{MLP}(\text{concat}(x_j, m_j^t)), s_j^{t-1}) \ .$$

The LSTM cell state is initialized to zeros.

The output function $r$ is a linear layer with nine outputs to produce the output logits $o_i^t$. We run the network for 32 steps with a loss on every step. We train the model for 300.000 gradient updates with a batch size of 256 using Adam with a learning rate of 2e-4 and L2 regularization of 1e-4 on all weight matrices.

## 1.6 Sudoku relational network baseline details

The node centric corresponds exactly to a single step of our network. The graph centric approach is closer to the original relational network. It does one step of relational reasoning as our network, then sums all the node hidden states. The sum is then passed through a 4 layer MLP with $81 \cdot 9$ outputs, one for each cell and digit. The graph centric model has larger hidden states of 256 in all layers to compensate somewhat for the sum squashing the entire graph into a fixed size vector. Otherwise both networks are identical to our network. The graph centric has over 4 times as many parameters as our model (944,874 vs. 201,194) but performs worse than the node centric.

## 1.7 Age arithmetic task details

We generated all 262,144 unique trees with 8 nodes and split them 90%/10% into training and test graphs. The nodes represent the persons, and the edges which age differences will be given to the network. During training and testing we sample a batch of graphs from the respective set and sample 8 random ages (0-99) for each. We compute the absolute difference as well as the sign for each edge in the graphs. This gives us 7 relative facts on the form "person A (0-7), person B (0-7), younger/older (-1,1), absolute age difference (0-99)". Then we add the final fact which is the age of one of the nodes at random, e.g. "3, 3, 0, 47", using the zero sign to indicate this fact is absolute and not relative. The question is the age of one of the persons at random (0-7). For each graph we compute the shortest path from the anchor person to the person in question. This is the minimum number of arithmetic computations that must be performed to infer the persons age from the given facts.

The 8 facts (1 anchor, 7 relative) are given to the network as a fully connected graph of 8 nodes. Note, this graph is different from the tree used to generate the facts. The network never sees the tree. The input vector for each fact are the four fact integers and the question integer one-hot encoded and concatenated. We use the same architecture as for the bAbI experiments except all MLPs are 3 dense layers with 128 ReLu units followed by one linear layer. We train the network for 8 steps, and test it for each step. See figure 1 for results.

## 1.8 Unrolled recurrent relational network

## 1.9 Full Sudoku solution

Figure 1: Results for the age arithmetic task. The number in parenthesis indicate how many steps the RRN was run during testing. Random corresponds to picking one of the 100 possible ages randomly.

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

Recurrent relational network on a fully connected graph with 3 nodes. Subscripts denote node indices and superscripts denote steps $t$. The dashed lines indicate the recurrent connections.

An example Sudoku. Each of the 81 cells contain each digit 1-9, which is useful if the reader wishes to try to solve the Sudoku as they can be crossed out or highlighted, etc. The digit font size corresponds to the probability our model assigns to each digit at step 0, i.e. before any steps are taken. Subsequent pages contains the Sudoku as it evolves with more steps of our model.

| | | | | | | | | |
|---|---|---|---|---|---|---|---|---|
| 4 | · | · | · | · | · | 6 | 9 | · |
| · | · | 3 | 2 | · | 4 | · | · | · |
| · | · | · | · | · | · | · | 1 | · |
| · | 3 / 9 | · | · | · | · | · | · | 2 |
| · | · | · | 8 | · | · | 5 | · | · |
| · | 1 | · | · | · | · | · | · | · |
| · | · | · | · | 9 | 1 | · | · | · |
| 8 | · | · | · | · | · | · | · | · |
| · | · | · | · | 6 | · | · | · | · |

(Cells marked "·" contain the pencil-mark grid of digits 1–9; cells with a large digit show the high-probability value(s) assigned by the model at step 0.)

Sudoku candidate grid (each cell shows its pencil-mark candidates 1–9; a single large digit denotes a solved cell):

| R\C | 1 | 2 | 3 | 4 | 5 | 6 | 7 | 8 | 9 |
|---|---|---|---|---|---|---|---|---|---|
| 1 | 4 | 2 5 7 8 | 1 2 5 7 8 | 1 3 5 7 | 1 3 5 7 8 | 3 5 7 8 | 6 | 5 9 | 3 5 7 8 |
| 2 | 1 5 6 7 9 | 5 6 7 8 9 | 3 | 2 | 1 5 7 8 | 4 | 7 8 | 5 7 8 | 5 7 8 |
| 3 | 2 5 6 7 9 | 2 5 6 7 8 9 | 2 5 6 7 8 | 3 7 9 | 3 5 7 8 | 3 5 6 7 8 9 | 2 3 4 7 8 | 1 | 3 4 5 7 8 |
| 4 | 5 6 7 | 3 9 | 9 | 1 4 5 6 7 | 1 4 5 7 | 5 6 7 | 1 4 7 8 | 4 6 7 8 | 2 |
| 5 | 2 7 | 2 4 6 7 | 2 4 6 7 | 8 | 1 2 3 4 7 | 2 3 6 7 9 | 5 | 3 4 6 7 | 1 3 4 6 7 9 |
| 6 | 2 5 6 7 | 1 | 2 4 5 6 7 8 | 3 4 5 6 7 9 | 2 3 4 5 7 | 2 3 5 6 7 9 | 4 7 8 9 | 3 4 6 7 8 | 3 4 6 7 8 9 |
| 7 | 2 3 5 6 7 | 2 4 5 6 7 | 2 4 5 6 7 | 3 4 5 7 | 9 | 1 | 2 3 4 7 8 | 2 3 4 5 6 7 8 | 3 4 5 6 7 8 |
| 8 | 8 | 2 4 5 6 7 9 | 1 2 4 5 6 7 | 3 4 5 7 | 2 3 4 5 7 | 2 3 5 7 | 1 2 3 4 7 | 2 3 4 5 6 7 | 1 3 4 5 6 7 9 |
| 9 | 1 2 3 5 7 9 | 2 4 5 7 9 | 1 2 4 5 7 | 3 4 5 7 | 6 | 2 3 5 7 8 | 1 2 3 4 7 8 9 | 2 3 4 5 7 8 | 1 3 4 5 7 8 9 |

Step 4

Step 8

A Sudoku candidate grid with pencil marks. Prominent (large) digits read per cell:

| | | | | | | | | |
|---|---|---|---|---|---|---|---|---|
| 4 | 2 | 1 | 5 / 7 | 5 / 8 | 5 / 7 | 6 | 9 | 3 |
| 6 / 9 | 3 | | 2 | 1 | 4 | 7 8 | 5 / 7 8 | 5 / 7 8 |
| 7 / 8 | 5 | | 6 / 9 | 3 | 6 / 9 | 2 | 1 | 4 |
| 5 6 / 7 | 3 | 9 | 1 | 4 5 / 7 | 5 6 / 7 | 4 / 7 8 | 4 6 / 7 8 | 2 |
| 2 / 6 / 7 | 4 / 7 | 2 / 4 6 / 7 | 8 | 2 / 7 | 3 / 9 | 5 | 3 6 / 7 | 1 / 9 |
| 2 5 6 / 7 | 1 | 8 | 3 6 / 9 | 2 4 5 3 / 7 9 | 2 3 5 6 / 9 | 3 4 7 9 | 3 4 6 7 8 | 6 / 7 9 |
| 3 | 4 5 6 / 7 | 2 4 6 / 7 | 4 5 / 7 | 9 | 1 | 4 7 8 | 2 4 5 6 / 7 8 | 6 / 7 8 |
| 8 | 4 5 6 / 7 9 | 1 2 4 5 6 / 7 | 3 4 5 / 7 | 2 5 / 7 | 2 3 | 1 3 | 2 3 4 5 6 / 7 9 | 1 5 6 / 7 9 |
| 1 | 2 4 5 / 7 9 | 2 4 / 7 | 3 4 5 / 7 | 6 / 8 | 2 3 / 8 9 | 3 4 5 / 7 9 | 2 3 4 5 / 7 8 | 5 / 7 9 |

Step 12

Step 16

A 9×9 Sudoku grid (Step 16) with given and pencil-mark digits.

Step 20

| | | | | | | | | |
|---|---|---|---|---|---|---|---|---|
| 4 | 2 | 1 | 7 | 8 | 5 | 6 | 9 | 3 |
| 9 | 6 | 3 | 2 | 1 | 4 | 7 | 8 | 5 |
| 7 | 8 | 5 | 9 | 3 | 6 | 2 | 1 | 4 |
| 5 | 3 | 9 | 1 | 4 | 7 | 8 | 6 | 2 |
| 6 | 4 | 7 | 8 | 2 | 9 | 5 | 3 | 1 |
| 2 | 1 | 8 | 6 | 5 | 3 | 9 | 4 | 7 |
| 3 | 7 | 6 | 5 | 9 | 1 | 4 | 2 | 8 |
| 8 | 9 | 4 | 3 | 7 | 2 | 1 | 5 | 6 |
| 1 | 5 | 2 | 4 | 6 | 8 | 3 | 7 | 9 |