[Reviews · NeurIPS 2018]

Reviewer 1



This paper presents a neural network module that operates on a graph to perform multiple steps of relational reasoning. The method is simple and general (in terms of you could choose other modules to compute the representation of the objects and also the node updates) and uses message passing for relational reasoning. My concerns are: 1.The proposed module and the message passing style computation is quite similar to other graph neural networks, so not sure this is very novel. 2.The evaluations are mostly done on synthetic datasets. The increase in performance on 6 and 7 steps in Figure 2b is concerning and indicates the flaws in the proposed "Pretty-CLEVR" dataset. 3.The evaluation on Sudoku is interesting. If you drop the position information and use only the edges and the number in the cell as input, how much would the performance drop? In my view, this makes the question more “relational”. 4.Although the model is general and in theory can be combined with complex neural modules. The representations in the evaluated domains are simple feature vectors. bAbI uses LSTM, but the language is quite simple and synthetic. It would be interesting to see how this module works with complex neural modules to deal with real images or natural language. -------------- I have updated the overall score from 5 to 6 since the author feedback addressed some of my concerns.

Reviewer 2



The paper proposed the recurrent relation network that takes into account multi step reasoning process. Along with the architecture, the authors also proposed the pretty CLEVER datatset which contains questions that needed to be answered via multi step reasoning. The authors experimented on the several different tasks including natural language Question Answering tasks (bAbI task) , Sudoku and the pretty clever dataset (multi-step Question Answer). The model performed better on all tasks and performed particularly well for Sudoku. Strength: 1. The multi-step reasoning task the model addresses is an important problem,. 2. The authors conducted thorough experiments and compared to many other baselines. 3 The authors provided thorough experimental settings in the appendix. Weakness: It would be interesting to see how this performs on tasks involving many-step reasoning with languages. Especially for tasks where the answer is not in the context given, but needs to be inferred. An example is "How old is person A?", the context is, Person B is 20 years old, Person C was born 4 years later and person A is 5 years older than person C. Post rebuttal. I am glad that authors found the suggested tasks useful and thanks for offering to acknowledge me in the final version, I would be happy to be acknowledges anonymously :) It is indeed interesting to see the results of this task. The model seems to perform significantly better compared to the baseline. It also serves as an interesting toy problem that analyzes the capability of the model. I am happy with the results presented and I am raising my scores to a 7 in light of the rebuttal.

Reviewer 3



This paper deals with designing a new neural network structure for problems involving one or more steps of relational reasoning. The authors propose a generalization of the Relational Network (RN) architecture proposed by Santoro 2017. Whereas RN does one step of processing on a fully-connected graph composed of nodes representing objects, the Recurrent Relational Network does multiple time steps of processing, maintaining a hidden state per node and parametrizing messages between nodes at each time step as a MLP. The new architecture is evaluated on several benchmarks. The first is bAbI, which is a popular text-based question-answering dataset consisting of 20 different type of tasks, each of which involves receiving several supporting facts and answering a question related to those facts. The proposed method solves all 20 tasks and seems to display less variability between different training runs in answering these questions compared to other published methods. The authors observe that bAbI is solved by their model without requiring multiple time-steps of processing in their model and so create a more difficult variant of CLEVR dataset (called Pretty-CLEVR), which is specifically built so that questions have varying degrees of reasoning. The authors evaluate their method, while varying the number of time steps of processing, and show that as they run the Recurrent Relational Network for more time steps, it is able to answer more difficult questions (in terms of more degrees of reasoning being required). Lastly, the method is evaluated on solving 9 x 9 Sudoku puzzles and achieves higher accuracy than a simple convolutional network and other graphical model based methods. Pros The proposed model has a simple definition but seems powerful based on its performance on various benchmarks. The Pretty-CLEVR experiment explicitly shows the benefit of allowing more time steps of processing with regard to questions that require more degrees of reasoning Cons Why not evaluate method on CLEVR dataset to see where method’s performance is relative to other published methods? Comments The model suggested in this work bears some similarity to the one used in Sainbayar 2016, which involves learning communication between multiple agents performing an RL-based task and the communication vector for one agent there is similarly summed across all other agents. Might be worth mentioning in the related works section. For the methods evaluated against in the Sudoku benchmark, are the results there from listed papers or based on implementation of those methods by the authors in order to compare? It would be good to make a note if any of the results are based on author’s implementation. Santoro, Adam et al. A simple neural network module for relational reasoning. 2017. Sukhbaatar, Sainbayar et al. Learning Multiagent Communication with Backpropagation. NIPS 2016.